# Belt Electrode-Skeletal Muscle Electrical Stimulation in Older Hemodialysis Patients with Reduced Physical Activity: A Randomized Controlled Pilot Study

**DOI:** 10.3390/jcm11206170

**Published:** 2022-10-19

**Authors:** Midori Homma, Misa Miura, Yo Hirayama, Tamao Takahashi, Takahiro Miura, Naoki Yoshida, Satoshi Miyata, Masahiro Kohzuki, Satoru Ebihara

**Affiliations:** 1Department of Internal Medicine & Rehabilitation Science, Disability Sciences, Tohoku University Graduate School of Medicine, 1-1 Seiryo-machi, Aoba-ku, Sendai 980-8574, Japan; 2Hirayama Hospital, Hanamigawa 1494-3, Hanamigawa-ku, Chiba 262-0046, Japan; 3Faculty of Health Sciences, Tsukuba University of Technology, Kasuga 4-12-7, Tsukuba 305-8521, Japan; 4Teikyo University Graduate School of Public Health, Kaga 2-11-1, Itabashi-ku, Tokyo 173-8605, Japan; 5Yamagata Prefectural University of Health Sciences, 260 Kamiyanagi, Yamagata 990-2212, Japan

**Keywords:** electrical muscle stimulation, hemodialysis, older, chronic kidney disease, frailty, 6 min walk test, short physical performance battery

## Abstract

Background: Although patients receiving hemodialysis are more likely to develop metabolic disorders and muscle weakness at an earlier stage than healthy individuals, many older dialysis patients have difficulty establishing exercise habits to prevent these problems. Therefore, we evaluated the use of belt electrode-skeletal muscle electrical stimulation (B-SES), which can stimulate a wider area than conventional electrical muscle stimulation (EMS), to examine its application and safety in older hemodialysis patients as a means to improve lower extremity function without voluntary effort. Methods: This study was a randomized controlled trial (RCT) involving 20 older dialysis patients (>65 years old) with reduced physical activity. The control group received 12 weeks of routine care only and the intervention group received 12 weeks of B-SES during hemodialysis in addition to routine care. The primary endpoint was the 6 min walk test (6MWT) distance, while the Short Physical Performance Battery (SPPB), body composition, Functional Independence Measure (FIM), biochemistry test, and blood pressure/pulse measurements were used as secondary endpoints. Results: As a result of the 12-week B-SES intervention, no increase in creatine kinase or C-reactive protein levels was observed after the intervention in either group, and no adverse events attributed to the B-SES intervention were observed in the intervention group. Furthermore, the intervention group showed a significant improvement in the 6MWT and SPPB scores after the intervention. Conclusions: The results of this study suggest that a 12-week B-SES intervention during hemodialysis sessions safely improves 6MWT distance and SPPB scores in older patients with a reduced level of physical activity.

## 1. Introduction

Hemodialysis patients tend to have increased oxidative stress and inflammatory responses due to uremia and cachexia during the preservation period of chronic kidney disease. Furthermore, as a result of decreased metabolic efficiency and lack of nutrients, hemodialysis patients tend to develop concomitant muscle wasting, cardiovascular disease, joint disease, and arteriosclerosis [1], and more than 70% of older hemodialysis patients are frail [2]. Furthermore, more than 30% of older hemodialysis patients also exhibit sarcopenia, with a prevalence approximately three times higher than that of healthy individuals [3]. Reduced exercise/physical function and a restricted level of physical activity are characteristics of individuals with frailty and sarcopenia and are associated with a decrease in survival rate, an increase in hospitalization rate, and a decrease in the quality of life (QOL) [4,5]. In particular, hemodialysis patients with less than 4000 steps of physical activity on days without hemodialysis sessions have a high mortality risk [6], and they are prone to a reduced tolerance to exercise, muscle strength, and muscle mass, and an increased cardiovascular risk [7,8]. As such, in Japan, where the number of older hemodialysis patients continues to increase, the importance of improving physical functions has been recognized so that older hemodialysis patients can continue to live independently.

In recent years, renal rehabilitation interventions focusing on supervised exercise therapy have been reported to improve muscle strength, exercise tolerance, and inflammation in hemodialysis patients or patients with renal failure [9,10], suggesting the impact of exercise on improved QOL. However, approximately 70% of older hemodialysis patients exhibit frailty, and many patients also have multiple disorders such as locomotor disorders and cardiovascular disorders, making it difficult for many of them to establish exercise habits [11,12]. In the clinical setting, it is often difficult to apply an effective exercise routine through active exercise. Furthermore, since hemodialysis patients receive hemodialysis treatments three times a week for 4 to 5 h per session, it is difficult for them to devote the remaining time to maintaining and improving physical function.

A recent study described the application of electrical muscle stimulation (EMS) using small self-adhesive surface electrode pads in patients with moderate to severe heart failure and respiratory disease and reported improvements in muscle strength, exercise tolerance, and walking ability [13]. Furthermore, improvements in muscle strength and exercise tolerance have also been reported in hemodialysis patients through local functional improvement dependent on surface electrodes [14]. However, in these interventions, the stimulation frequency/time, intervention period, equipment used, electrode pad size, and electrical stimulation intensity have yet to be standardized; thus, supporting evidence is limited [14]. The belt electrode-skeletal muscle electrical stimulation (B-SES) device has a wider electrode attachment area than conventional EMS and is expected to be more effective in improving lower limb muscle strength and the Short Physical Performance Battery (SPPB) score in patients with orthopedic diseases and patients with severe heart failure [15,16].

Furthermore, in recent years, B-SES intervention during hemodialysis sessions has been reported to improve muscle strength, muscle mass, and Timed Up & Go test scores without adverse events in middle-aged patients [17]. However, to date, no B-SES interventions have been conducted in older hemodialysis patients who are more frail and vulnerable to adverse events. Additionally, they are more prone to muscle disuse due to immobility. Therefore, in this study, we hypothesized that B-SES intervention during hemodialysis would improve physical function without adverse events in older hemodialysis patients. Thus, our objective was to investigate the comprehensive impact of B-SES on lower limb function using the 6 min walk test (6MWT) and SPPB as outcome measures.

## 2. Materials and Methods

### 2.1. Study Design

This study was a prospective open label randomized controlled trial (RCT). Patients were enrolled at a single center and data analysis was performed in multiple centers. At the start of the study, medical records were screened, and patients were enrolled and divided into an intervention group or a control group. A permuted block design, generated by RAND function in Microsoft Excel, was used to evenly randomize the patients into the two groups. We then collected demographic data from the patients and performed baseline evaluations one week before the intervention. The following variables were evaluated at baseline: physical function (as assessed by the 6MWT and SPPB), post-dialysis weight (dry weight), body composition (skeletal muscle mass of the extremities), activities of daily living (ADL), blood biochemistry, and blood pressure/pulse. Hypertension was diagnosed if, when measured on two different days, both systolic blood pressure (SBP) readings were ≥140 mmHg and/or both diastolic blood pressure readings (DBP) were ≥90 mmHg. Dyslipidemia was diagnosed if low-density lipoprotein cholesterol levels were ≥100 mg/dL, high-density lipoprotein cholesterol levels were >40 mg/dL, or triglyceride levels were ≥150 mg/dL. Diabetes mellitus was diagnosed if fasting blood glucose levels were ≥126 mg/dL. We also collected data concerning history of ischemic heart diseases and cerebrovascular diseases from the patients’ medical records.

After completion of the 12-week intervention, the same variables were measured again within 1 week (Figure 1). The study period ran from 24 June 2020 to 31 March 2022, and the patient enrollment period was from 24 June 2020 to 31 December 2021.

A note was made in the study notebook of all events that occurred during the intervention. Blood pressure and pulse were measured before the start of each hemodialysis session using the blood pressure monitor of a multipurpose hemodialysis monitoring device (Nikkiso DCS-100NX, Tokyo, Japan).

This study was conducted following approval by the Tohoku University Clinical Research Review Committee (reference number: 2019-6-063) and registration in the Ministry of Health, Labour and Welfare data system (jRCT Protocol No.: jRCTs022200010) and was conducted in accordance with the Declaration of Helsinki. The patients were individually informed of the study rationale using an informed consent form prior to enrollment, and only those who provided their consent were enrolled in the study. The results of this study were reported according to the 2010 CONSORT Guidelines [18].

### 2.2. Participants

The inclusion criteria for the study were as follows: (1) patients with end-stage kidney disease that had been introduced to hemodialysis, (2) patients who had been receiving hemodialysis for more than 3 months, (3) patients with stable hemodynamics receiving outpatient maintenance hemodialysis, (4) patients aged between 65 and 90 years old at the time of enrollment, (5) patients with daily activity level of less than 4000 steps [6], and (6) patients who were briefed about the study and provided voluntary written consent to participate.

The exclusion criteria were as follows: (1) patients with sensory impairment, (2) patients who had difficulty walking, (3) patients with malignant neoplasms, (4) patients with severe edema, (5) patients with severe skin diseases or wounds on the belt-type electrode attachment site, (6) patients with systolic blood pressure of 180 mmHg or higher or diastolic blood pressure of 110 mmHg or higher, (7) patients participating in other clinical studies, (8) patients requiring acute treatment for acute coronary syndrome, unstable angina, or other conditions, (9) patients with implantable electronic devices (such as pacemakers), (10) patients undergoing temporary pacing or intra-aortic balloon pumping therapy, etc., (11) patients with psychiatric disorders or severe dementia, and (12) patients deemed ineligible for the study by the investigator for other reasons.

### 2.3. Intervention Protocol

B-SES was performed using G-TES (Homer Ion Institute Co., Ltd., Tokyo, Japan), which is an electrical stimulator for general treatment (Figure 2). The stimulation was carried out in “Obsolete mode” (frequency: 20 Hz, on-off: 5 sec–2 sec, pulse width: 250 μs, output waveform: exponentially increasing wave) for 40 min per session, according to the median time reported in a previous study by Schardong et al. [14]. All other stimulation conditions were in accordance with a previous study by Suzuki et al. [17].

The intensity of stimulation was the maximum intensity tolerated by the patient. The intervention period was 40 min per day, 3 times a week, for 12 weeks, totaling 36 sessions. Belt-type electrodes were attached to a total of five locations in the trunk, both thighs, and both lower legs of the patients, and electrical stimulation was performed in the first half of the hemodialysis time.

Stimulation intensity was adjusted at two locations, the thigh and lower leg. The median/mean intensity during the first intervention was 1.70/1.79 (minimum 0.8–maximum 2.8) mA at the thigh and 0.95/0.97 (0.4–1.4) mA at the lower leg. At the final intervention, it was 3.15/3.47 (2.1–7.4) mA at the thigh and 1.55/1.74 (1.1–3.9) mA at the lower leg.

Patients in both the control group and the intervention group were instructed to lead the same lifestyle as before the intervention, and no special exercise instructions were given during the intervention period. After the intervention period, all patients underwent voluntary training guidance.

### 2.4. Outcome Measures

All evaluation variables were measured at the beginning of the study and at the end of the 12-week intervention (Figure 1).

#### 2.4.1. Physical Function Tests

In this study, the 6MWT distance (6MWD), an indicator of exercise intolerance, was the primary endpoint. The 6MWD was assessed using a 15 m one-way corridor and all assessment procedures were performed according to the guidelines of the American Thoracic Society (ATS) [19].

A comprehensive assessment of lower extremity function was performed using the Short Physical Performance Battery (SPPB), according to a previous study [20]. The assessment procedure consisted of (1) tandem stand, (2) 4 m walk test, and (3) 5-Time-Sit-To-Stand (5-STS) tests, and the times and scores were recorded. For all measurement elements, the use of knee braces and walking aids was optional.

#### 2.4.2. Body Compositions

Measurements were taken by a physical therapist after hemodialysis sessions using a body composition analyzer (Tanita Corporation MC-780A-N) [21]. The measurement items were dry weight and skeletal muscle mass in the extremities.

#### 2.4.3. ADL Assessment

ADL were measured by a physiotherapist through an interview survey using the Functional Independence Measure (FIM) [22].

#### 2.4.4. Biochemical Parameters

Blood biochemistry tests were performed during the baseline assessment and on the last day of the intervention. Blood biochemistry tests examined creatine kinase (CK), blood urea nitrogen (BUN), C-reactive protein (CRP), interleukin-6 (IL-6), insulin-like growth factor 1 (IGF-1), total antioxidant capacity (TAC), and irisin levels. In addition, using BUN values before and after hemodialysis, the hemodialysis efficiency (spKt/V) was calculated as spKt/V = –ln (post-dialysis BUN/pre-dialysis BUN − 0.008 * dialysis time) + (4 − 3.5 * post-dialysis BUN/pre-dialysis BUN) * amount of water removed/DW [23].

### 2.5. Statistical Analysis

All analyses were two-sided with a significance level of 5%. The statistical analysis software was SPSS ver.21 (IBM Corp., Chicago, IL, USA).

Demographic and laboratory data were expressed as mean ± standard deviation or median (interquartile range) for continuous variables, and as number of people (%) for categorical variables. The Shapiro–Wilk test was performed for normality of distribution. For baseline comparisons between the control and intervention groups, continuous variables were analyzed using the unpaired t-test or the Mann–Whitney U test, and categorical variables were analyzed using the chi-square test. For comparison of pre- and post-intervention results between groups, all parameters were analyzed using the paired t-test or Wilcoxon signed rank sum test. For comparisons between the control group and the intervention group after 12 weeks, the unpaired t-test or the Mann–Whitney U test was used. Furthermore, differences in the amount of change between the control and intervention groups were assessed using the unpaired t-test or the Mann–Whitney U test.

The sample size required for the paired analysis of 6MWD was calculated based on a previous EMS study [24], which used a significance level of 5%, a power of 80%, and an effect size of 0.5, resulting in a minimum requirement of 13 participants in each group.

## 3. Results

A total of 27 older hemodialysis patients were enrolled from 69 candidate participants who met the inclusion criteria and did not violate the exclusion criteria. The 27 patients were randomized into the control group (13 patients) or the intervention group (14 patients) (Figure 3). Seven patients dropped out. Of the four patients who withdrew from the intervention group, two patients were admitted to other hospitals and were unavailable for the endpoint assessment of 6MWD, one patient refused to participate in the endpoint assessment, and one patient had a non-measurable 6MWD due to a thigh injury. Of the three patients who withdrew from the control group, one patient was admitted to another hospital and was unavailable for the 6MWD measurement, and two patients refused to participate in the endpoint assessment.

Therefore, we used a per protocol analysis. A total of 20 patients completed the intervention (control group: 10 patients, intervention group: 10 patients), and their data were analyzed.

Table 1 shows the intergroup comparisons of demographic data at baseline. There were no significant differences between the two groups in terms of age (control group: 78.4 ± 6.2 years old, intervention group: 79.4 ± 6.5 years old), sex ratio (control group: 60.0% male, intervention group: 70.0% male), number of steps (control group: 796.25 (427.50–3276.19) steps/day, intervention group: 1525.38 (735.50–3454.31) steps/day), history of hemodialysis (control group: 34.00 (26.75–50.75) months, intervention group: 28.50 (12.25–79.25) months), or prevalence of diabetes (control group: 40.0%, intervention group: 70.0%).

Table 2 shows the blood pressure and pulse measurements. There were no significant differences in SBP, DBP, or pulse.

Table 3 shows the results of 6MWD, the primary endpoint. In the control group, no significant differences were observed between the measurements at baseline and 12 weeks. However, in the intervention group, there was a significant increase in 6MWD from 242.0 ± 94.5 m to 283.0 ± 99.6 m (*p* = 0.005). In addition, there was a significant difference between the amount of change in 6MWD between the two groups, with −18.50 ± 28.29 m in the control group and 41.00 ± 12.65 m in the intervention group (*p* < 0.001).

Table 3 shows the SPPB results. In the control group, there was no significant difference in the total SPPB score at baseline and 12 weeks. However, there was a significant improvement in the intervention group from 8.5 ± 3.3 points to 10.3 ± 2.1 points (*p* = 0.008). There was also a significant difference in the amount of change in SPPB between the two groups (*p* = 0.035). There were no significant differences in dry weight, skeletal muscle mass in the extremities, or FIM (Table 3).

Table 4 shows the results of the blood biochemistry tests. Although there was a significant decrease in albumin levels from baseline to 12 weeks in the control group (*p* = 0.027), this change was not observed in the intervention group and there were no significant differences between the amount of change in albumin levels between the two groups. There were no significant differences in the levels of CK, irisin, IGF-1, TAC, CRP, IL-6, hemoglobin, or in dialysis efficiency.

No locomotor difficulties or cardiovascular events were reported in this study. Four patients in the intervention group withdrew from the study to be hospitalized or withdrew their consent to continue in the study. None of the withdrawals were attributable to effects resulting from the participation in the study.

## 4. Discussion

Many older hemodialysis patients have sarcopenia [3], which has been reported to be associated with decreased physical function, an increased hospitalization rate, and decreased QOL and ADL [10,12]. Therefore, continuous exercise intervention is necessary, but many hemodialysis patients have difficulty establishing exercise habits given the time constraints of hemodialysis treatment, and it is often difficult for older hemodialysis patients with reduced physical function to continue voluntary exercise at an effective load. Therefore, the purpose of this study was to examine the effects of B-SES in older patients on maintenance hemodialysis from the perspective of safety and efficacy in improving physical function.

In this study, to examine the safety of the B-SES intervention in older patients with hemodialysis, we measured CK, CRP, IL-6, blood pressure, and pulse before hemodialysis and verified changes before and after the intervention. To evaluate the effects of electrical stimulation on muscle damage and inflammation, we measured CK, CRP, and IL-6 levels and observed no changes in any parameter before or after the intervention in either group. An increase in CRP and a decrease in albumin have been reported to cause inflammation in the body and contribute to a decrease in muscle mass [25], but in this study, despite the drop in albumin levels in the control group, no changes were observed in albumin levels or skeletal muscle mass of the extremities in the intervention group. No additional inflammatory disease or pneumonia occurred during the intervention, nor did any of the patients change drugs that could affect inflammation, such as antibiotics. Furthermore, neither group showed changes in blood pressure and pulse after the intervention. These observations suggest that B-SES can be implemented for older hemodialysis patients without serious adverse events.

This study is the first to show that a B-SES intervention in older hemodialysis patients with reduced physical activity improved 6MWD, an indirect measure of exercise tolerance, and SPPB scores, an index of comprehensive lower extremity function. A strong correlation between 6MWD and SPPB has been found in previous studies targeting older patients with heart disease and older patients in general [26,27]. Furthermore, SPPB is strongly correlated with lower extremity muscle strength [20], and B-SES interventions have been reported to strengthen lower extremity muscles in previous studies on orthopedic surgery and middle-aged hemodialysis patients [15,17].

Moreover, the improvement in 6MWD and SPPB scores, despite the absence of an increase in muscle mass, may be attributed to their relationship to lower extremity muscle strength. Increases in muscle strength tend to occur earlier than increases in muscle mass, as higher intensity exercise is required to increase muscle mass [28]. Since general EMS passively induces muscle contraction through electrical stimulation via surface electrodes, it has been suggested that even patients who have difficulty moving can achieve exercise effects [14]. Muscle contraction by EMS differs from muscle contraction by general voluntary contraction in that fast-twitch fibers with large fiber diameters (Type II fibers) are mobilized first and muscle contraction is induced only in areas that receive electrical stimulation [29]. In EMS, slow-twitch fibers and fast-twitch fibers capable of continuous contraction (Type II A fibers) have been reported to be predominantly activated by low-frequency stimulation of 20 Hz or less [30]. Increases in muscle strength and muscle mass are often correlated in general resistance training, but there is very little evidence of the effect of EMS on increasing muscle mass [13]. Since the intensity of electrical stimulation was low in this study, the intensity of exercise was weaker than that of resistance training and cannot be regarded as a sufficient load to increase muscle mass. However, we believe that muscle fibers were activated by stimulating the trunk and leg muscles with B-SES, resulting in increased muscle strength. These findings suggest that B-SES improved lower extremity muscle strength as well as 6MWD and SPPB scores.

According to reports in patients with chronic obstructive pulmonary disease (COPD), patients with chronic heart failure, and older patients, the minimal clinically important difference (MCID) of 6MWD ranges from 17 to 54 m [31,32]. In the present study, the change in 6MWD in the intervention group was 41.0 ± 12.0 m, thus falling within this MCID range. Additionally, the change in the total SPPB score in the intervention group was 1.8 ± 1.7 points, exceeding the MCID range of 0.99 to 1.34 points for total SPPB score in older patients [33,34]. These findings show that the B-SES intervention in older hemodialysis patients resulted in clinically significant improvements in both 6MWD and SPPB.

In this study, B-SES intervention for older hemodialysis patients (mean age 78 years) was safe and sustainable. Although there were no changes in muscle mass and blood biochemical parameters, there were improvements in 6MWD and SPPB scores, which may have been as a result of the selective activation of the trunk and leg muscles through electrical stimulation. Dobsak et al. conducted a three-group comparison of EMS vs. exercise training on an ergometer vs. no exercise (control) in middle-aged hemodialysis patients [35]. They found that when comparing the EMS group and the voluntary exercise groups via resistance training and exercise with an ergometer, voluntary exercise had a greater effect on improving physical function. On the basis of the above, B-SES may be useful as a complementary exercise method for older people and patients who have difficulty maintaining voluntary exercise routines.

This study has two limitations, the first being the small sample size. Therefore, we positioned this study as a pilot study. The second is the lack of measurements of muscle strength. In the future, it would be necessary to examine the effects of B-SES on muscle strength by increasing the number of participants and measuring muscle strength.

## 5. Conclusions

In this study, a 12-week B-SES intervention was performed safely without serious adverse events during hemodialysis sessions in older patients. Furthermore, 6MWD and SPPB scores improved post-intervention, suggesting that B-SES may improve physical function in older people with decreased physical activity.

## Figures and Tables

**Figure 1 jcm-11-06170-f001:**
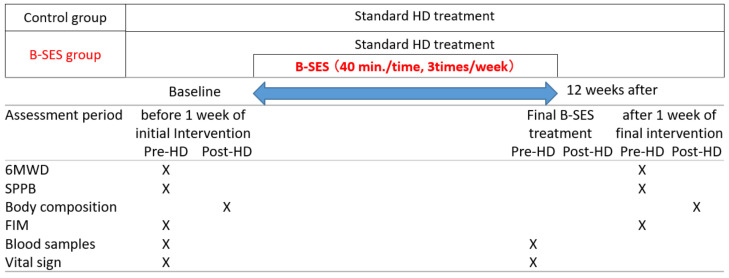
Schematic diagram of study schedule. Most variables were assessed pre-hemodialysis, while body composition was assessed post-hemodialysis. “Vital sign” included assessment of systolic and diastolic blood pressure. 6MWD, 6-Minute Walk Distance; SPPB, Short Physical Performance Battery; and FIM, Functional Independence Measurement.

**Figure 2 jcm-11-06170-f002:**
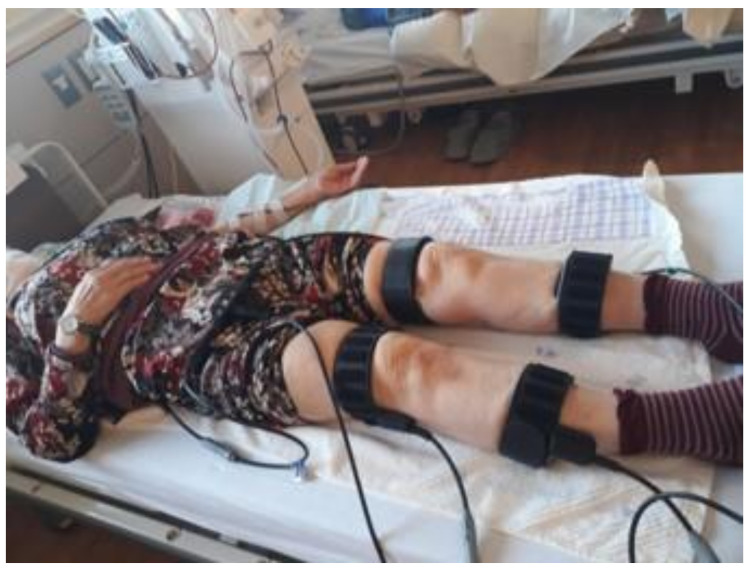
Application of the belt electrode-skeletal muscle electrical stimulation intervention. Patients were placed in the supine position. Five silicon–rubber electrode bands were applied to the patient’s waist and bilateral thighs and ankles. Electrical muscle stimulation was administered for 40 min per day.

**Figure 3 jcm-11-06170-f003:**
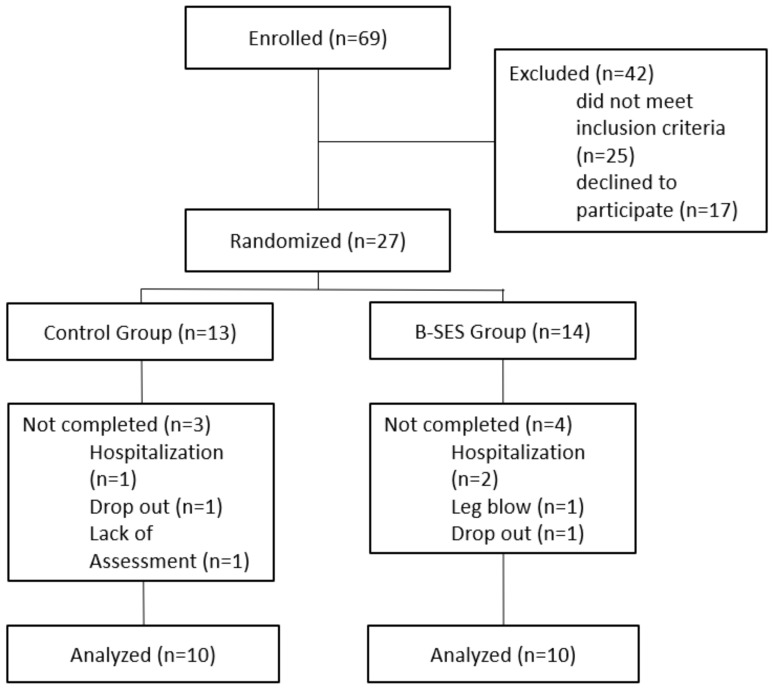
Study flow chart. Among the 69 patients who were screened for eligibility, 42 patients were excluded because of their physical function/medical assessment results or because they declined. Therefore, a total of 20 patients (10 in the control group and 10 in the B-SES group) were included in the final analysis.

**Table 1 jcm-11-06170-t001:** Clinical characteristics.

	Control Group (n = 10)	B-SES Group (n = 10)	*p* Value
Age (years)	78.40 ± 6.20	79.40 ± 6.50	0.73 ^a^
Men, n (%)	6 (60.00)	7 (70.00)	1.00 ^c^
Height (cm)	158.11 ± 8.87	154.25 ± 6.49	0.28 ^a^
Body composition			
Dry weight (kg)	50.92 ± 12.00	52.35 ± 3.78	0.73 ^a^
Skeletomuscular mass (kg)	15.83 ± 3.99	15.05 ± 2.40	0.62 ^a^
Physical activity (steps/day)	796.25 [427.50–3276.19]	1525.38 [735.50–3454.31]	0.44 ^b^
Duration of hemodialysis (months)	34.00 [26.75–50.75]	28.50 [12.25–79.25]	0.63 ^b^
Not completed rate, n (%)	3 (23.08)	4 (28.57)	1.00 ^c^
Comorbidity			
Hypertension, n (%)	9 (90)	9 (90)	1.00 ^c^
Dyslipidemia, n (%)	1 (10)	4 (40)	0.30 ^c^
Diabetes, n (%)	4 (40)	7 (70)	0.37 ^c^
History of ischemic heart disease, n (%)	7 (70)	7 (70)	1.00 ^c^
History of cerebrovascular disease, n (%)	4 (40)	8 (80)	0.17 ^c^
Medication			
Erythropoietin, n (%)	7 (70)	7 (70)	1.00 ^c^
L-carnitine, n (%)	9 (90)	8 (80)	1.00 ^c^

Values are presented as mean ± standard deviation or median (inter-quartile range (IQR)) unless otherwise indicated; Significance level *p* < 0.05; Baseline, before 1 week of initial intervention; ^a^ analyzed using the unpaired t-test; ^b^ analyzed using the Mann–Whitney U test; ^c^ analyzed using the chi-square test.

**Table 2 jcm-11-06170-t002:** Resting vital signs.

	Control Group (n = 10)			B-SES Group (n = 10)			Change from Baseline		
	Baseline	12 Weeks	*p* Value	Baseline	12 Weeks	*p* Value	Control Group	B-SES Group	*p* Value
SBP (mmHg)	153.00 ± 20.36	148.50 ± 21.03	0.21	151.00 ± 23.13	147.70 ± 24.21	0.71	−3.40 ± 10.98	−7.10 ± 15.09	0.54
DBP (mmHg)	74.90 ± 11.57	74.40 ± 10.03	0.76	72.40 ± 16.75	71.00 ± 10.36	0.75	−2.30 ± 11.16	2.90 ± 9.93	0.29
Pulse (bpm)	70.30 ± 13.48	66.40 ± 11.57	0.22	72.50 ± 12.07	75.30 ± 8.15	0.31	1.60 ± 8.34	2.30 ± 7.44	0.85

Values are presented as mean ± standard deviation or median (IQR); Significance level *p* < 0.05; Baseline, before 1 week of initial intervention; SBP, Systolic Blood Pressure; and DBP, Diastolic Blood Pressure.

**Table 3 jcm-11-06170-t003:** Physical assessments.

	Control Group (n = 10)			B-SES Group (n = 10)			Change from Baseline		
	Baseline	12 Weeks	*p* Value	Baseline	12 Weeks	*p* Value	Control Group	B-SES Group	*p* Value
Physical function test									
6MWD (m)	271.00 ± 90.88	252.50 ± 88.98	0.07	242.00 ± 94.49	283.00 ± 99.62	<0.01	−18.50 ± 28.29	41.00 ± 12.65	<0.001
SPPB Total (score)	8.80 ± 2.53	9.20 ± 2.78	0.10	8.50 ± 3.34	10.30 ± 2.11	0.01	0.40 ± 0.70	1.80 ± 1.81	0.04
SPPB Tandem (score)	2.50 [2.00–4.00]	3.00 [3.00–4.00]	0.18	3.00 [2.75–3.25]	4.00 [3.00–4.00]	0.02	0.00 [0.00–1.00]	1.00 [0.00–1.25]	0.25
SPPB Gait (score)	4.00 [3.00–4.00]	4.00 [3.00–4.00]	0.32	3.00 [2.50–4.00]	3.40 [2.71–4.00]	0.10	0.00 [0.00–0.00]	0.00 [0.00–1.00]	0.17
SPPB 5-STS (score)	2.50 [1.00–3.25]	3.00 [0.75–4.00]	0.48	3.00 [1.00–4.00]	3.50 [2.75–4.00]	0.06	0.00 [−0.25–1.00]	0.00 [0.00–1.00]	0.44
Body composition									
Dry weight (kg)	50.92 ± 12.00	51.49 ± 11.47	0.29	52.35 ± 3.78	52.31 ± 4.18	0.87	0.57 ± 1.59	0.04 ± 0.73	0.29
Skeletomuscular muss (kg)	15.83 ± 3.99	16.05 ± 3.90	0.38	15.05 ± 2.40	15.13 ± 2.53	0.74	0.22 ± 0.76	0.08 ± 0.75	0.68
ADL									
FIM (score)	121.50 [116.75–123.25]	120.50 [117.25–124.00]	0.72	121.00 [118.50–123.00]	121.00 [113.75–123.25]	0.50	0.00 [−3.00–1.50]	0.00 [−2.25–1.00]	0.91

Values are presented as mean ± standard deviation or median (IQR); Significance level *p* < 0.05; Baseline, before 1 week of initial intervention; 6MWD, 6 min walk distance; SPPB, Short Physical Performance Battery; 5-STS, 5-Time-Sit-To-Stand; ADL, activity of daily living; and FIM, functional independence measurement.

**Table 4 jcm-11-06170-t004:** Biochemical parameters.

	Control Group (n = 10)			B-SES Group (n = 10)			Change from Baseline		
	Baseline	12 Weeks After	*p* Value	Baseline	12 Weeks After	*p* Value	Control Group	B-SES Group	*p* Value
CK (U/L)	81.70 ± 36.99	83.10 ± 50.10	0.93	74.90 ± 28.65	86.80 ± 45.47	0.23	1.40 ± 50.64	11.90 ± 29.32	0.58
Irisin (ng/mL)	6.81 [6.16–7.91]	5.95 [5.68–6.86]	0.14	6.24 [5.78–8.12]	7.00 [5.68–7.82]	0.59	−0.21 [−1.27–1.10]	−0.89 [−2.65–0.29]	0.35
TAC (mM)	2.15 ± 0.49	2.63 ± 0.78	0.13	2.03 ± 0.53	1.82 ± 1.05	0.54	0.49 ± 0.93	−0.21 ± 1.06	0.13
CRP (mg/dL)	0.14 [0.10–0.28]	0.09 [0.05–0.12]	0.19	0.26 [0.06–0.38]	0.13 [0.05–0.51]	0.86	−0.50 [−0.23–0.02]	−0.01 [−0.13–0.23]	0.32
IL-6 (pg/mL)	3.05 ± 1.67	2.58 ± 1.17	0.44	3.87 ± 3.38	3.06 ± 2.15	0.44	−0.48 ± 1.86	−0.81 ± 3.15	0.78
IGF-1 (ng/mL)	84.20 ± 35.56	95.40 ± 41.16	0.27	78.40 ± 18.66	77.70 ± 28.15	0.94	11.20 ± 30.06	−0.70 ± 27.69	0.37
Albumin (mg/dL)	3.58 ± 0.30	3.46 ± 0.33	0.02	3.69 ± 0.26	3.60 ± 0.23	0.24	−0.12 ± 0.14	−0.09 ± 0.22	0.72
Hemoglobin (mg/dL)	9.95 ± 0.96	10.08 ± 0.55	0.53	10.14 ± 1.10	9.92 ± 0.82	0.47	0.13 ± 0.63	−0.22 ± 0.93	0.34
spKt/V	1.46 ± 0.25	1.55 ± 0.26	0.29	1.48 ± 0.28	1.66 ± 0.31	0.05	0.09 ± 0.27	0.18 ± 0.25	0.47

Values are presented as mean ± standard deviation or median (IQR); Significance level *p* < 0.05; Baseline, before 1 week of initial intervention; CK, Creatine Kinase; TAC, Total Antioxidant Capacity; CRP, C-reactive protein; IL-6, interloikin-6; and IGF-1, Insulin-like growth factors-1.

## Data Availability

The data presented in this study are available on request from the corresponding author.

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
