# Peer review of "Belt Electrode-Skeletal Muscle Electrical Stimulation in Older Hemodialysis Patients with Reduced Physical Activity: A Randomized Controlled Pilot Study"

_jcm, 2022, doi:10.3390/jcm11206170_

Round 1

Reviewer 1 Report

The authors investigate an important issue facing ESKD patients and have designed a trial to assess if the use of belt electrode skeletal muscle electrical stimulation can improve physical function in older ESKD patients. Overall I commend the authors on designing an RCT to answer their research question and below are a few comments to the authors that I hope they can find helpful to improve this study. 

Why did the authors just focus on the older population >65. Why not include all age groups to see if there is benefit with earlier intervention?

As this was an open label trial and both the patient and investigator knew which arm the patient was assigned to, it is important to describe how patients were randomized. Was it permuted block randomization for example?

There is also no mention of whether this was an intention to treat analysis. Did all patients complete follow up (4 withdrew in the intervention group). How was loss to follow up or withdrawal from the study dealt with ?

How was the number 27 decided on? Was there a power calculation?

In terms of practicality. How expensive is this belt. Is it something than can be given to every dialysis patient. 

Lastly there are many spelling errors that the authors should pay attention to; for example creating instead of creatinine in the abstract, supervised instead of supervised.

Author Response

Responses to Reviewer 1’s comments

We thank you for the time you have invested in reviewing our manuscript and for the favorable comments. For ease of reading, we have organized our responses below by providing the Comment (C) followed by our Response (R). The changes in the revised version of the manuscript are shown using Track Changes functions in Microsoft Word. Please note that the changes made do not influence the content, conclusions, or framework of the paper.

C1: The authors investigate an important issue facing ESKD patients and have designed a trial to assess if the use of belt electrode skeletal muscle electrical stimulation can improve physical function in older ESKD patients. Overall I commend the authors on designing an RCT to answer their research question and below are a few comments to the authors that I hope they can find helpful to improve this study.

R1: Thank you for your constructive feedback. We have carefully considered your comments and have revised the manuscript accordingly.

C2: Why did the authors just focus on the older population >65. Why not include all age groups to see if there is benefit with earlier intervention?

R2: Unlike older dialysis patients, young dialysis patients tend to be more active in their daily lives and rarely experience issues associated with muscle disuse due to immobility. Therefore, we focused on older dialysis patients in this study (please refer to line 86-86 in the revised manuscript).

C3: As this was an open label trial and both the patient and investigator knew which arm the patient was assigned to, it is important to describe how patients were randomized. Was it permuted block randomization for example?

R3: Yes, it was. A permuted block design, generated by RAND function in Microsoft Excel, was used to evenly randomize the patients into the two groups. We have included this description of this randomization process in the methods (please refer to line 98-100 in the revised manuscript).

C4: There is also no mention of whether this was an intention to treat analysis. Did all patients complete follow up (4 withdrew in the intervention group). How was loss to follow up or withdrawal from the study dealt with ?)

R4: Thank you for pointing out this important issue. In our study, the primary endpoint was the evaluation of the distance in the 6-minute walk test (6MWD) at 12 weeks from baseline. evaluation.

Of the four patients who withdrew from the intervention group, two patients were admitted to other hospitals and were unavailable for the endpoint assessment, one patient refused to participate in the endpoint assessment, and one patient had a non-measurable 6MWD due to thigh injury.

Of the three patients who withdrew from the control group, one patient was admitted to another hospital at the time of endpoint evaluation and was unavailable for the 6MWD measurement, and two patients refused to participate in the endpoint assessment.

Therefore, we used a per protocol analysis.

We have added this information in the revised manuscript (please refer to line 242-252).

C5: How was the number 27 decided on? Was there a power calculation?

R5: The sample size required for the paired analysis of 6MWD was calculated based on a previous study (reference #24), which used a significance level of 5%, a power of 80%, an effect size of 0.5, resulting in a minimum requirement of 13 participants in each group. We have added this description in the manuscript (please refer to line 234-237).

C6: In terms of practicality. How expensive is this belt. Is it something than can be given to every dialysis patient.

R6: The current price of the belt electrode (B-SES) set, G-TES (Homer Ion Institute Co., Ltd., Tokyo, Japan), is 1,500,000 yen (approximately 10,000 US$). Unfortunately, the use of B-SES is not currently reimbursed by the Japanese health insurance system. We hope that with the accumulation of evidence, the use of B-SES will be covered by health insurance in the future.

C7: Lastly there are many spelling errors that the authors should pay attention to; for example creating instead of creatinine in the abstract, supervised instead of supervised.

R7: Thank you for your feedback. We have carefully checked the spelling throughout the manuscript. In addition, we have asked Editage to review the English throughout the manuscript.

Reviewer 2 Report

The study described results of Belt Electrode Skeletal Muscle Electrical Stimulation (B-SES) on clinical and laboratory parameters. It is a well-prepared study without major methodological errors. The manuscript is clearly written.

Major remarks:

The main limitation of these randomized study is the small number of patients who completed the study (10 in the intervention and control group)

Minor remarks:

Line 54 Word „suoervised” please correct.

Figure 1. Vital sign – please provide the description.

Line 149. How was measured “The intensity of stimulation”? Please provide  the lowest, medial/mean and highest values.

Line 209. “ratio” or sex ratio ?

Table 1. Please provide definitions for Hypertension, Dyslipidemia, Diabetes, Ischemic heart disease, Cerebrovascular disease in Methods section.

Table 1. Please provide the name of the statistical test for assessing intergroup differences.

Line 303. Please correct the sentence “Since the intensity of the intensity”

Line 332. The "handheld dynamometer" is not a suitable instrument for measuring the strength of the lower limbs!

Question:

Were the control subjects doing any exercise or receiving any advice on increasing physical activity during the study?

Author Response

Reviewer 2`s comments

We appreciate the time you have invested in reviewing our manuscript and we thank you for your constructive comments. For ease of reading, we have organized our responses below by providing the Comment (C) followed by our Response (R). The changes in the revised version of the manuscript are shown using Track Changes functions in Microsoft Word. Please note that the changes made do not influence the content, conclusions, or framework of the paper.

C1: The study described results of Belt Electrode Skeletal Muscle Electrical Stimulation (B-SES) on clinical and laboratory parameters. It is a well-prepared study without major methodological errors. The manuscript is clearly written.

R1: Thank you for your considerate feedback. We have revised the manuscript according to your comments.

Major remarks:

C2: The main limitation of these randomized study is the small number of patients who completed the study (10 in the intervention and control group).

R2: The sample size required for the paired analysis of 6MWD was calculated based on a previous study (reference #24), which used a significance level of 5%, a power of 80%, an effect size of 0.5, resulting in a minimum requirement of 13 participants in each group (line 234-235 in the revised manuscript). Although we recruited 14 patients for the intervention group and 13 patients for the control group, the final sample size for analysis was 20 (10 in each group). As you pointed out, our sample size was small, but our results were significant. Therefore, we have rewritten the title as “A Randomized Controlled Pilot Study” (line 3-4 in the revised manuscript) and have referred to this in the discussion regarding sample size issue (please refer to line 389-390 in the revised manuscript).

Minor remarks:

C3: Line 54 Word „suoervised” please correct.

R3: Thank you for bringing this to our attention. We have corrected this error (please refer to line 56 in the revised manuscript).

C4: Figure 1. Vital sign – please provide the description.

R4: Thank you for bringing this to our attention. “Vital sign” included assessment of systolic and diastolic blood pressure. We have provided this description in the caption for Figure 1 (please refer to line 155-156 in the revised manuscript).

C5: Line 149. How was measured “The intensity of stimulation”? Please provide the lowest, medial/mean and highest values.

R5: Stimulation intensity was adjusted at two locations, the thigh and the lower leg. The median/mean intensity during the first intervention was 1.70/1.79 (minimum 0.8 - maximum 2.8) mA at the thigh and 0.95/0.97 (0.4-1.4) mA at the lower leg. At the final intervention it was 3.15/3.47 (2.1-7.4) mA at the thigh and 1.55/1.74 (1.1-3.9) mA at the lower leg.

We have added this description in the manuscript (please refer to line 176-180).

C6: Line 209. “ratio” or sex ratio ?

R6: Thank you for the clarification. We have corrected this to “sex ratio” (please refer to line 257 in the revised manuscript).

C7: Table 1. Please provide definitions for Hypertension, Dyslipidemia, Diabetes, Ischemic heart disease, Cerebrovascular disease in Methods section.

R7: Hypertension was diagnosed if, when measured on two different days, both systolic blood pressure readings were ≥140 mmHg and/or both diastolic blood pressure readings were ≥90 mmHg.

Dyslipidemia was diagnosed if 1) LDL cholesterol levels were ≥100 mg/dL, 2) HDL cholesterol levels were > 40 mg/dL, or 3) triglyceride levels were ≥150 mg/dL.

Diabetes mellitus was diagnosed if fasting blood glucose levels were ≥126 mg/dL.

The diagnoses of ischemic heart disease and cerebrovascular disease were confirmed from the patient’s medical history record.

We have added these descriptions (please refer to line 106-114) and corrected Table 1 in the revised manuscript.

C8: Table 1. Please provide the name of the statistical test for assessing intergroup differences.

R8: We apologize for this lack of the information in the previous manuscript. We have provided the name and details of the statistical tests for assessing intergroup differences (indicated by superscript letters beside the p-values) in the footnote of Table 1 (please refer to line 273-275 in the revised manuscript).

C9: Line 303. Please correct the sentence “Since the intensity of the intensity”

R9: Thank you for bringing this error to our attention. We have corrected this sentence (please refer to line 358-359 in the revised manuscript).

C10: Line 332. The "handheld dynamometer" is not a suitable instrument for measuring the strength of the lower limbs!

R10: We completely agree with your opinion. Accordingly, we have deleted "handheld dynamometer" from the manuscript (please refer to line 393 in the revised manuscript).

Question:

C11: Were the control subjects doing any exercise or receiving any advice on increasing physical activity during the study?

R11: Patients in both the control group and the intervention group were instructed to lead the same lifestyle as before the intervention, and no special exercise instructions were given during the intervention period. After the intervention period ended, all patients underwent voluntary training guidance.

We have added this description to the manuscript (please refer to line 181-184).

Round 2

Reviewer 1 Report

Thank you for the replies and revision

This sufficiently answers all the concerns that were brought up by initial review. No further comments.